# Single molecule analysis reveals reversible and irreversible steps during spliceosome activation

Aaron A Hoskins[1]*, Margaret L Rodgers[1], Larry J Friedman[2], Jeff Gelles[2], Melissa J Moore[3]

[1]Department of Biochemistry, University of Wisconsin-Madison, Madison, United States; [2]Department of Biochemistry, Brandeis University, Waltham, United States; [3]Department of Biochemistry and Molecular Pharmacology, Howard Hughes Medical Institute, University of Massachusetts Medical School, Worcester, United States

**Abstract** The spliceosome is a complex machine composed of small nuclear ribonucleoproteins (snRNPs) and accessory proteins that excises introns from pre-mRNAs. After assembly the spliceosome is activated for catalysis by rearrangement of subunits to form an active site. How this rearrangement is coordinated is not well-understood. During activation, U4 must be released to allow U6 conformational change, while Prp19 complex (NTC) recruitment is essential for stabilizing the active site. We used multi-wavelength colocalization single molecule spectroscopy to directly observe the key events in *Saccharomyces cerevisiae* spliceosome activation. Following binding of the U4/U6.U5 tri-snRNP, the spliceosome either reverses assembly by discarding tri-snRNP or proceeds to activation by irreversible U4 loss. The major pathway for NTC recruitment occurs after U4 release. ATP stimulates both the competing U4 release and tri-snRNP discard processes. The data reveal the activation mechanism and show that overall splicing efficiency may be maintained through repeated rounds of disassembly and tri-snRNP reassociation.

*For correspondence: ahoskins@ wisc.edu

**Competing interests:** The authors declare that no competing interests exist.

## Introduction

The spliceosome is one of the most dynamic molecular machines inside the cell. Removal of introns from precursors to mRNAs (pre-mRNAs) involves the coordinated action of 5 small nuclear RNAs (snRNAs) and >100 proteins (*Wahl et al., 2009*; *Hoskins and Moore 2012*). Some of these proteins along with the snRNAs assemble into small nuclear ribonucleoprotein particles (the U1, U2, U4, U5, and U6 snRNPs) that work together with other accessory proteins to catalyze splicing. Experiments in vitro (*Hoskins et al., 2011*; *Ruby and Abelson 1988*; *Konarska and Sharp 1986*; *Cheng and Abelson 1987*) and in cells (*Tardiff and Rosbash 2006*) indicate that spliceosomes are unlikely to exist as preformed complexes. Instead, spliceosomes are built from their snRNP and accessory protein components on pre-mRNAs, carry out splicing, and then are disassembled after each reaction (*Wahl et al., 2009*). Consequently, the overall process can be described as sequential progression through distinct stages of spliceosome assembly, formation of the active site (called activation), catalysis, disassembly, and component recycling. A number of biochemical and genetic experiments have elucidated the splicing factors present at each stage (*Wahl et al., 2009*; *Fabrizio et al., 2009*), as well as characteristic interactions between snRNA, pre-mRNA, and protein components (*Brow 2002*; *Wahl et al., 2009*).

The U1 and U2 snRNPs identify the 5' splice site (SS) and branch site (BS), respectively, during early stages of spliceosome assembly. While U1 binding is ATP-independent, U2 base pairing with

**eLife digest** The genes in an organism's DNA may be expressed to form a protein via an intermediate molecule called RNA. In many organisms including humans, gene expression often begins by making a precursor molecule called a pre-mRNA. The pre-mRNA contains regions called exons that code for the protein product and regions called introns that do not. A machine in the cell called the spliceosome has the job of removing the introns in the pre-mRNA and stitching the exons together by a process known as splicing.

The spliceosome is made up of dozens of components that assemble on the pre-mRNAs. Before a newly assembled spliceosome can carry out splicing, it must be activated. The activation process involves several steps that are powered by the cell's universal power source (a molecule called ATP), including the release of many components from the spliceosome. Many of the details of the activation process are unclear.

Spliceosomes in the yeast species *Saccharomyces cerevisiae* are similar to spliceosomes from humans, and so are often studied experimentally. Hoskins et al. have now used a technique called colocalization single molecule fluorescence spectroscopy to follow, in real time, a single yeast spliceosome molecule as it activates. This technique uses a specialized microscope and a number of colored lasers to detect different spliceosome proteins at the same time. Hoskins et al. found that one of the steps during activation is irreversible – once that step occurs, the spliceosome must either perform the next activation steps or start the processes of assembly and activation over again.

Hoskins et al. also discovered that ATP causes some spliceosomes to be discarded during activation and not used for splicing. This indicates that before spliceosomes are allowed to activate, they may undergo 'quality control', which may be important for making sure that gene expression occurs efficiently and correctly. Future studies will investigate how this quality control process works in further detail.

the intron to form the pre-spliceosome or A complex typically requires ATP hydrolysis (*Figure 1A*). A pre-formed complex of U4, U5, and U6 (the U4/U6.U5 tri-snRNP) then joins A complex to form B complex. While B complex contains dozens of proteins and five snRNAs, it is not capable of mediating either of the two chemical steps of splicing (5' SS cleavage and exon ligation) since the spliceosomal components are not yet rearranged into a configuration capable of catalysis. This rearrangement encompasses two stages during the activation process. In the first stage, the Prp19-associated complex (NTC), the final major spliceosomal subcomplex, joins and U1 and U4 are expelled to form the $B^{act}$ spliceosome. In the second stage, $B^{act}$ is further remodeled to the $B^*$ complex (*Lardelli et al., 2010*; *Wlodaver and Staley 2014*; *Liu and Cheng 2012*). Single molecule FRET (smFRET) experiments suggest that it is only in B* complex that the 5' SS and BS become juxtaposed, a necessary prerequisite for formation of a spliceosome competent for 5' SS cleavage (*Crawford et al., 2013*; *Krishnan et al., 2013*). These catalytically activated $B^*$ spliceosomes then progress further through stages of exon ligation, mRNA product release, and finally disassembly of the lariat intron-containing product complex.

Despite the central importance of the B-to-$B^{act}$ transition, key mechanistic questions remain unanswered. For example, the order of U4 loss and NTC association is uncertain and alternative models have been proposed (*Fabrizio et al., 2009*; *Chan et al., 2003*; *Tarn et al., 1993*). It is also unclear whether U4 snRNP loss is irreversible or whether it can rejoin an activated spliceosome to fix incorrectly assembled complexes by possible proofreading or spliceosomal discard pathways.

Single-molecule fluorescence microscopy techniques have proven valuable in elucidating the kinetic mechanisms of several parts of the splicing process (*Hoskins et al., 2011*; *Abelson et al., 2010*; *Shcherbakova et al., 2013*; *Crawford et al., 2013*; *Krishnan et al., 2013*; *Cherny et al., 2010*; *Karunatilaka and Rueda 2013*; *Guo et al., 2009*). To investigate the mechanism of spliceosomal activation, we have used Colocalization Single Molecule Spectroscopy (CoSMoS) (*Friedman et al., 2006*; *Hoskins et al., 2011*) to study the binding dynamics of U4 relative to other spliceosome components. CoSMoS is particularly suited for the study of unsynchronized reactions such as spliceosomal activation since individual reaction trajectories can be followed in real time and

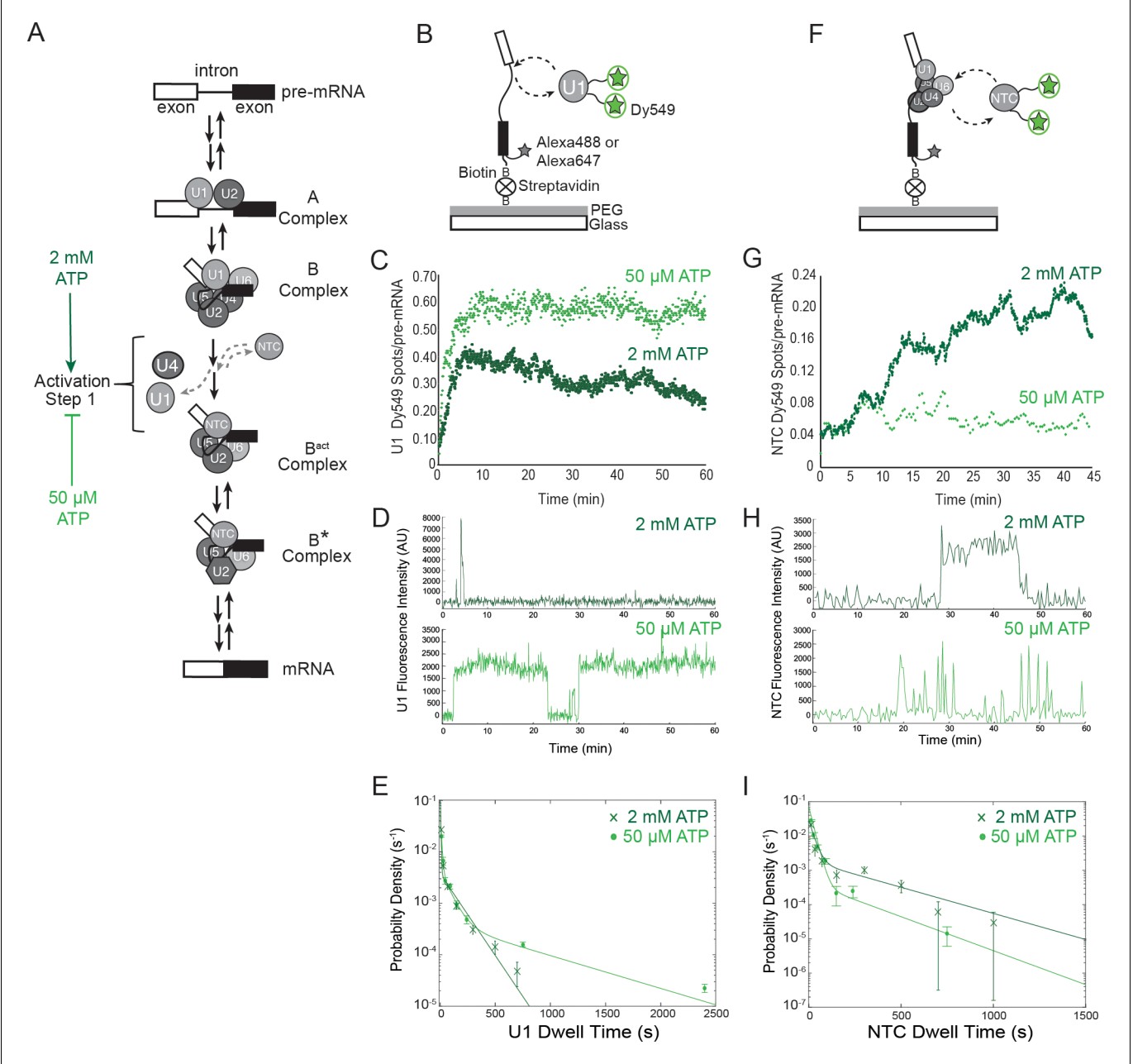

**Figure 1.** Cartoon of major steps in spliceosome assembly and activation and impact of 2 mM (dark green) and 50 µM (light green) ATP concentrations on U1 and NTC interactions with pre-mRNA. (**A**) Spliceosome assembly and activation. snRNPs interact with the pre-mRNA to form the A complex, containing U1 and U2 associated with the 5' SS and BS, respectively. The U4/U6.U5 tri-snRNP is then recruited to form B complex. At 2 mM ATP, activation can proceed and result in release of U1 and U4 and acquisition of the NTC to form B$^{act}$. Subsequent steps then lead to splicing. At 50 µM ATP, the first step of activation is inhibited. (**B**) Schematic of a two-color CoSMoS experiment for observing U1 binding dynamics. The U1 snRNP contained two green-excited (Dy549) fluorophores attached to two different proteins while the pre-mRNA was immobilized to the slide surface and contained a single Alexa 488 or Alexa 647 fluorophore. (**C**) Time record of the number of U1 fluorescence spots relative to the number of surface-tethered pre-mRNA molecules at 2 mM and 50 µM ATP. (**D**) Example fluorescence intensity records showing binding of U1 to individual pre-mRNA molecules at 2 mM and 50 µM ATP. (**E**) Probability density histogram of dwell times for U1 at 2 mM and 50 µM ATP (N = 535 events on 166 pre-mRNAs at 50 µM ATP). Results at 2 mM ATP were originally reported by Hoskins et al. and those data sets were used to generate this new figure for comparison (*Hoskins et al., 2011*). Lines represent fits of the distributions of dwell times to multi-exponential equations. (**F**) Schematic of a two-color CoSMoS experiment for observing NTC binding dynamics. (**G**) Time record of the number NTC fluorescence spots relative to the number of surface-tethered pre-mRNA molecules at 2 mM and 50 µM ATP. (**H**) Example fluorescence intensity records showing binding of the NTC to individual pre-mRNA molecules at 2 mM and 50 µM ATP. (**I**) Probability density histogram of dwell times for the NTC at 2 mM and 50 µM ATP (N = 234 events on 169 pre-mRNAs at 50 µM ATP). Results at 2 mM ATP were originally reported by Hoskins et al. and those data sets were used to generate this new figure

*Figure 1 continued on next page*

*Figure 1 continued*

for comparison (*Hoskins et al., 2011*). Lines represent fits of the distributions of dwell times to multi-exponential equations. Parameters for fits in (**E**) and (**I**) are shown in *Figure 1— source data 1*.

The following source data and figure supplements are available for figure 1:

**Source data 1.** Fit parameters describing the distribution of dwell times observed for the U1 and NTC subcomplexes.

**Figure supplement 1.** Additional examples (to supplement those shown in *Figure 1D*) of fluorescence intensity traces showing individual U1-SNAP subcomplexes co-localizing with surface-tethered pre-mRNA in the presence of 2 mM (**A–E**, dark green) or 50 µM (**F–J**, light green) ATP.

**Figure supplement 2.** Additional examples (to supplement those shown in *Figure 1H*) of fluorescence intensity traces showing individual NTC-SNAP subcomplexes co-localizing with surface-tethered pre-mRNA in the presence of 2 mM (**A–E**, dark green) or 50 µM (**F–J**, light green) ATP.

subsequently grouped into common patterns of behavior during data analysis (*Larson et al., 2014*; *Walter et al., 2008*; *Chen et al., 2012*). Here we followed the comings and goings of U4, U5 and the NTC under conditions that permitted or prevented spliceosomal activation. By simultaneously observing multiple species, we were able to distinguish reaction pathways relevant to activation from off-pathway processes and define the order and reversibility of on-pathway binding and dissociation steps. Together the data reveal new features of spliceosomal activation, provide evidence for an ATP-dependent checkpoint that results in rapid tri-snRNP discard, and define the major pathway for NTC recruitment.

## Results

### Limiting ATP changes U1 and NTC dynamics

In previous studies, we have shown that snRNPs and the NTC can dynamically engage pre-mRNAs under ATP concentrations that permit splicing (2 mM) (*Hoskins et al., 2011*; *Shcherbakova et al., 2013*). In the absence of ATP, however, only U1 snRNP could stably interact with pre-mRNA (*Hoskins et al., 2011*), consistent with previous spliceosome assembly models (*Legrain et al., 1988*; *Wahl et al., 2009*). In between these two extremes, lower amounts of ATP (*e.g.*, 50 µM) allow for spliceosome assembly up to tri-snRNP addition (B complex formation) but prevent loss of U4 snRNP and spliceosomal activation (*Figure 1A*) (*Chan et al., 2003*; *Tarn et al., 1993*). This has proven particularly useful for isolating B complex and investigating its composition (*Fabrizio et al., 2009*). We wondered if a similar approach could be used to stall spliceosomes to observe B complex on single pre-mRNA molecules and gain insights into the mechanisms of spliceosomal activation in single molecule experiments.

Based on the abundance of U1 snRNP in biochemically-purified, stalled B complex (*Fabrizio et al., 2009*), we predicted that these stalled spliceosomes could be detected at the single molecule level by an increase in U1 snRNP dwell time at 50 µM ATP compared to dwell times measured under conditions that permit U1 release and splicing. We previously described procedures for monitoring U1 snRNP interactions with surface-tethered pre-mRNAs in yeast whole cell extract (WCE) using green-laser excitable (Dy549) SNAP tag fluorophores on U1 components (Snp1, Prp40; *Table 1*) and RP51A pre-mRNA containing a 5' cap, 3' biotin handle, and a single, site-specifically incorporated Alexa647 (red-laser excitable) or Alexa488 (blue-laser excitable) fluorophore (*Figure 1B*) (*Hoskins et al., 2011*).

As expected, U1 snRNPs initially bound RP51A pre-mRNAs at both 2 mM and 50 µM ATP (*Figure 1C*). However, the profile of U1 fluorescent spot accumulation under the two conditions was quite different: at 2 mM ATP, U1 snRNP spot numbers increased, then decreased as spliceosomes underwent activation; at 50 µM ATP, U1 spots reached a higher surface density that did not decrease with time (*Figure 1C*). The latter result is consistent with U1 remaining bound to pre-activation spliceosomal B complexes. Inspection of individual U1 binding events on single pre-mRNAs confirmed this assumption. In 50 µM ATP, U1 spots often remained visible for tens of minutes (e.g., *Figure 1D*, light green and *Figure 1—figure supplement 1*). This is much longer than previously observed either in the absence of ATP (which prevents A complex formation) or at an ATP

**Table 1.** Yeast strains and labeled proteins.

| Strain | SNAP Tag Labels[a] | DHFR Tag Labels | Genotype | References |
|---|---|---|---|---|
| yAAH0001 BJ2168 | None | None | MATa prc1–407 prb1–1122 pep4–3 leu2 trp1 ura3–52 gal2 | Parental Strain |
| yAAH0006 | U1: Snp1, Prp40 | None | yAAH001 + snp1::snp1-SNAP-HYG + prp40::prp40-SNAP-NAT | *Hoskins et al., 2011* |
| yAAH0020 | NTC: Cef1, Ntc90 | None | yAAH001 + cef1::cef1-SNAP-HYG + ntc90::ntc90-SNAP-NAT | *Hoskins et al., 2011* |
| yAAH0067 | U4: Prp3, Prp4 | None | yAAH001 + prp3::prp3-SNAPf-Hyg + prp4::prp4-SNAPf-NAT | this work |
| yAAH0071 | U4: Prp3 | U5: Brr2, Snu114 | yAAH001 + prp3::prp3-SNAPf-NAT + brr2::brr2-DHFR-HYG + snu114::snu114-DHFR-BLE | this work |
| yAAH0077 | U4: Prp3 | NTC: Cef1, Ntc90 | yAAH001 + prp3::prp3-SNAPf-NAT + cef1::cef1-DHFR-HYG + ntc90::ntc90-DHFR-BLE | this work |

[a]Strains yAAH0067, 71, and 77 were labeled with the fast SNAP (SNAP$_f$) tag

concentration (2 mM) that supports splicing (*Figure 1D*, dark green and *Figure 1—figure supplement 1*; [*Hoskins et al., 2011*]). The distribution of U1 dwell times fit a three component model. This model contained short and intermediate dwell time components (~0.15 and 1.3 min, respectively) previously observed in 2 mM ATP (*Hoskins et al., 2011*) plus a much longer-lived component (≥11 min) new to the 50 µM ATP condition (*Figure 1E* and *Figure 1—source data 1*). It is likely that these long-lived U1s are in stalled B complexes analogous to those previously purified from 50 µM ATP splicing reactions (*Fabrizio et al., 2009*; *Tarn et al., 1993*; *Chan et al., 2003*).

It has been previously found that the NTC does not co-purify with B complex but only with activated spliceosomes (*Tarn et al., 1993*; *Chan et al., 2003*; *de Almeida and O'Keefe 2015*). To investigate whether and for how long NTC associates with stalled B complexes, we used yeast extracts bearing green-excited SNAP tag fluorophores on NTC components (Cef1 and Ntc90) to compare NTC association with pre-mRNA at 2 mM and 50 µM ATP (*Figure 1F*, *Table 1*). As previously observed (*Hoskins et al., 2011*), spots of NTC fluorescence accumulated on a surface with tethered pre-mRNAs in the presence of 2 mM ATP. However, at 50 µM ATP we observed much less NTC accumulation (*Figure 1G*). These results suggest that either the NTC cannot associate with B complex or it can associate only transiently, consistent with previous hypotheses (*Tarn et al., 1993*; *Chan et al., 2003*). Inspection of individual pre-mRNAs showed fewer bound NTC molecules in 50 µM ATP and that NTC binding events often lasted for only a few seconds (*Figure 1H*, light green and *Figure 1—figure supplement 2*). Analysis of the observed dwell times showed that at both 2 mM and 50 µM ATP, NTC binding events could be described by a distribution containing two exponential terms corresponding to short (~0.4 min) and long (~5 min) dwell time components (*Figure 1I* and *Figure 1—source data 1*). However, under 50 µM ATP the long dwell time component represented only a negligible fraction of the distribution (amplitude of 0.06 vs. 0.46 for 50 µM and 2 mM ATP, respectively). Thus, the kinetically more stable interactions of NTC that are present at 2 mM ATP are largely absent at 50 µM ATP. In summary, the single-molecule U1 and NTC binding dynamics under low ATP conditions are consistent with the formation of stalled, B complex spliceosomes on surface tethered pre-mRNAs.

## Limiting ATP results in long-lived U4 binding events

To gain further information about the step at which spliceosome activation is blocked at 50 µM ATP, we incorporated fluorophores into U4/U6-snRNP proteins (Prp3 and Prp4, [*Hardin et al., 2015*]) that were previously shown to be lost during activation along with the U4 snRNA (*Fabrizio et al., 2009*). We used homologous recombination to create a yeast strain harboring SNAP$_f$ tags (*Sun et al., 2011*) on both Prp3 and Prp4 (*Table 1* and *Figure 2—source data 1*). This strain was viable, had a similar growth rate to the untagged parental strain (*Figure 2—figure supplement 1A*), and yielded WCE in which both SNAP$_f$ tagged proteins could be specifically and rapidly labeled (*Figure 2—figure supplement 1B,C*). Further, the labeled extract was active in bulk splicing reactions (*Figure 2—figure supplement 1D,E*).

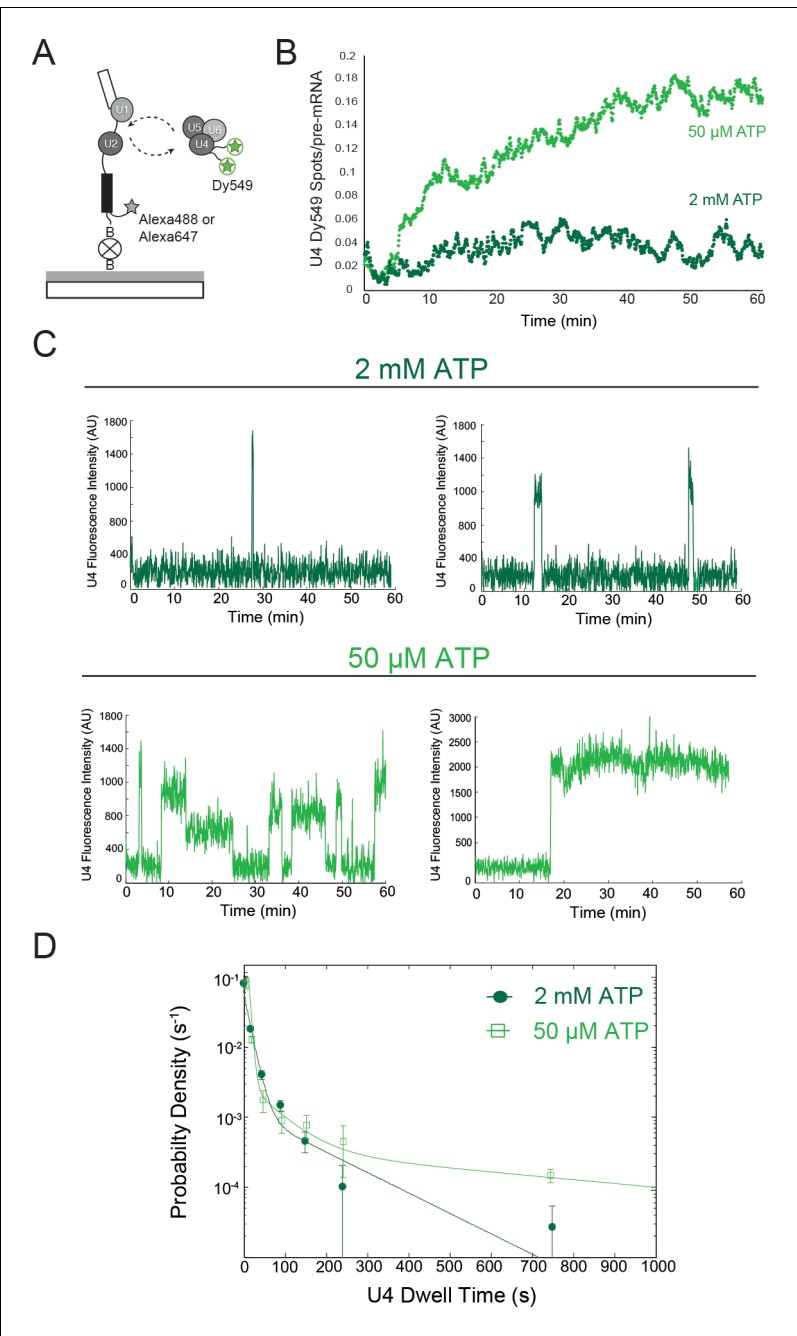

**Figure 2.** Two-color CoSMoS observation of U4 dynamics at 2 mM (dark green) and 50 µM (light green) ATP. (**A**) Schematic of an experiment for observing U4 binding dynamics with a design similar to that in *Figures 1B* and 1F. (**B**) Time record of the number U4 fluorescence spots relative to the number of surface-tethered pre-mRNA molecules at 2 mM and 50 µM ATP. (**C**) Example fluorescence intensity records showing binding events of U4 to single pre-mRNA molecules at 2 mM and 50 µM ATP. (**D**) Probability density histogram of dwell times for the U4 at 2 mM (*N* = 336 events on 229 pre-mRNAs) and 50 µM ATP (*N* = 151 events on 226 pre-mRNAs). Lines represent fits of the distributions of dwell times to multi-exponential equations; fit parameters for U4 are given in *Figure 2— source data 2*.

The following source data and figure supplements are available for figure 2:

**Source data 1.** Oligonucleotides used for generating yeast strains.

**Source data 2.** Fit parameters describing the distribution of dwell times observed for the U4 subcomplex.

*Figure 2 continued on next page*

*Figure 2 continued*

**Figure supplement 1.** Characterization of U4-labeled yeast strains and splicing extracts.

**Figure supplement 2.** Additional examples (to supplement those shown in *Figure 2C*) of fluorescence intensity traces showing individual U4-SNAP subcomplexes co-localizing with surface-tethered pre-mRNA in the presence of 2 mM (**A–E**, dark green) or 50 µM (**F–J**, light green) ATP.

We performed two-color CoSMoS experiments to detect U4 interactions with surface-tethered pre-mRNAs via the labels on Prp3 and Prp4 (*Figure 2A*; *Video 1*). At 50 µM ATP, U4 signals accumulated on the surface and reached a high density that did not decrease with time (*Figure 2B*, light green), similar to the behavior we observed with U1. At 2 mM ATP we saw little accumulation of U4 signals (*Figure 2B*, dark green). As for U1, these results suggest that at limiting ATP, U4 remains bound to pre-activation spliceosomal B complexes. However, we do not yet know if complexes formed at limiting ATP that contain U4 are the same complexes that also contain U1. The properties of individual U4 binding events on single pre-mRNAs were consistent with B complex formation. In 50 µM ATP, some U4 particles interacted transiently while others remained associated for tens of minutes, whereas at 2 mM ATP nearly all U4 binding was transient (*Figure 2C*; *Figure 2—figure supplement 2*). Also consistent with these observations, at 2 mM ATP U4 exhibited two comparatively short lifetime components of ~0.2 and ~2 min (*Figure 2D*; *Figure 2—source data 2*). In contrast, at 50 µM ATP the measured dwell times fit to a tri-exponential distribution with two similar short components (~0.1 min and ~1 min), but with a substantial fraction in an additional much longer component (≥13 min). We hypothesize that this long component arises from the formation of stalled B complex spliceosomes at 50 µM ATP. Absence of this component at 2 mM ATP is consistent with the idea that binding of U4/U6.U5 to form B complex is followed quickly by activation and concomitant U4 dissociation.

## Three-color CoSMoS identifies distinct U4 and U5 arrival and departure pathways

In WCE, U4 snRNP exists in three major complexes: U4 snRNP alone, the U4/U6 di-snRNP, and the U4/U6.U5 tri-snRNP (*Raghunathan and Guthrie 1998a*). Although it is presumed that only the tri-snRNP binds to the U1.U2 pre-spliceosome, this has not been directly tested. Therefore, it is possible that the short-lived dwell time components reported above might represent transient interactions of U4 alone or U4/U6 di-snRNP with pre-spliceosomes. In order to specifically identify tri-snRNP binding events, we constructed a yeast strain bearing labels on both U5 and the U4/U6 component Prp3 (*Table 1*). This strain grew similarly to the parental strain and its WCE exhibited high levels of in vitro splicing activity (*Figure 2—figure supplement 1*). We then used the WCE in a three color CoSMoS experiment with red-excited DHFR tags on U5 (Brr2, Snu114), a green-excited SNAP$_f$ tag reporting on U4 (Prp3), and blue-excited RP51A pre-mRNA (*Figure 3A*, *Video 2*).

At 2 mM ATP, three-color CoSMoS experiments showed that the largest cohort of U5 spot arrivals (63%; 551 of 876 total) on pre-mRNA molecules exhibited simultaneous appearance of

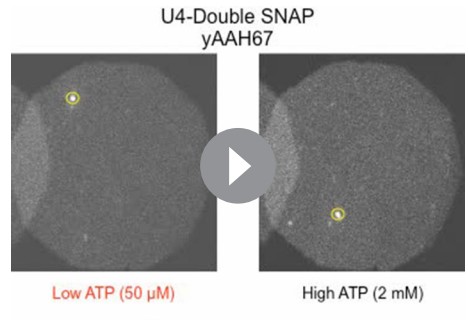

**Video 1.** Surface accumulation of the U4 snRNP on surface-tethered pre-mRNAs (not shown) under both low (50 µM) and high (2 mM) ATP conditions. Spliceosome activation is prevented at low ATP. White spots represent fluorescent U4 subcomplexes labeled with Dy549-SNAP$_f$ tags. This movie was obtained from recordings (60 min) of 1 s duration frames recorded at 3 s intervals. To reduce the movie file size, only every third frame was included from 0 to 45 min in the movie file, and the movie is played back at 20 frames/s. Each of the two recordings shows a ~20 x 20 µm field of view, and the two individual movie files were combined using Apple Keynote software.

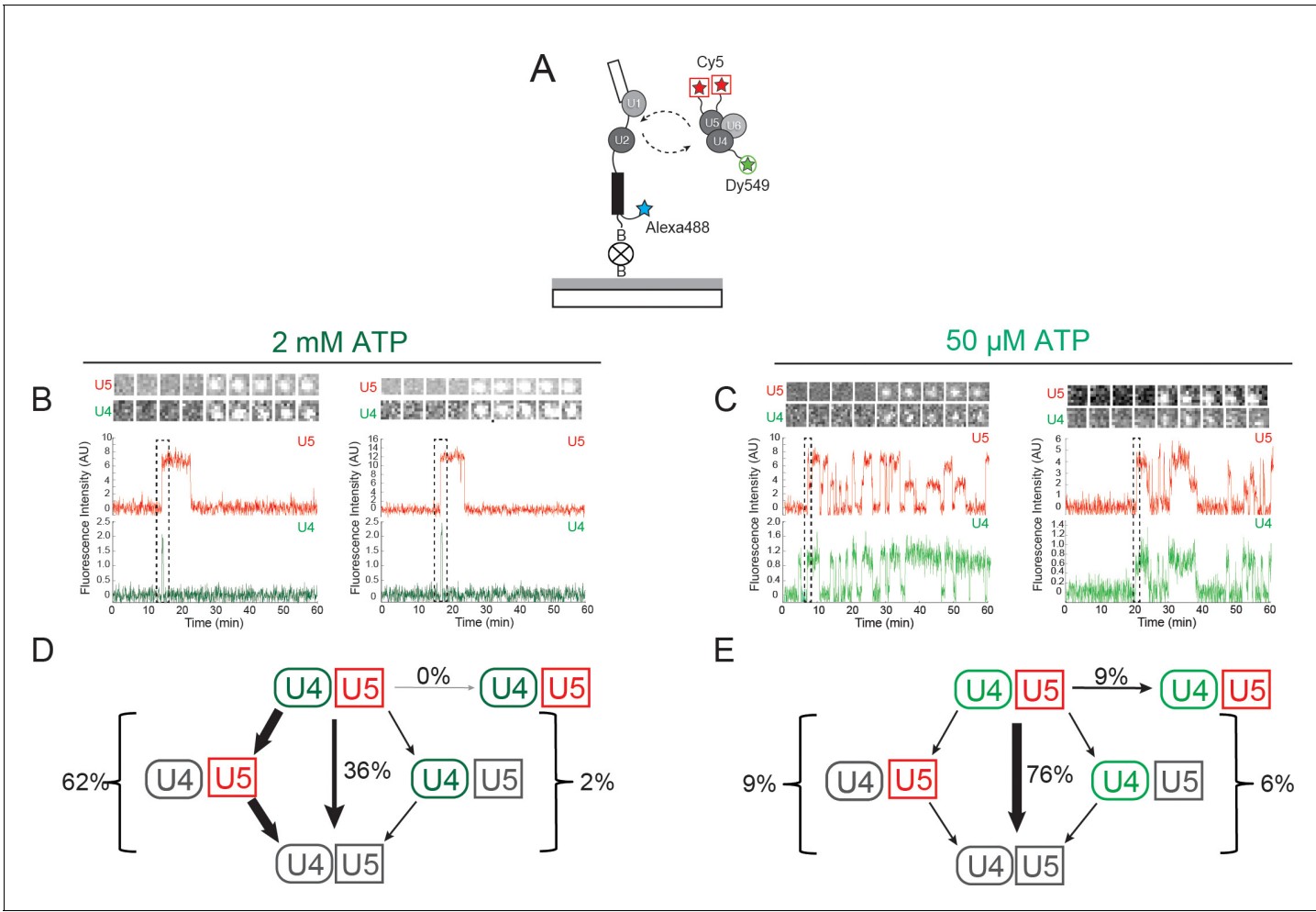

**Figure 3.** Three-color CoSMoS observation of U4 and U5 binding dynamics at 2 mM and 50 μM ATP. (**A**) Schematic of a three-color experiment in which U5 was labeled with two red-excited (Cy5) fluorophores, U4 was labeled with a single green-excited (Dy549) fluorophore, and the surface-tethered pre-mRNA was labeled with a single blue-excited (Alexa488) fluorophore. (**B** and **C**) Representative time records at 2 mM (**B**) or 50 μM ATP (**C**), each showing peaks in fluorescence intensity corresponding to colocalization of U4 (green) and U5 (red) with the same individual pre-mRNA molecule. Dashed rectangles mark examples of the simultaneous appearance of U4 and U5 spots; galleries show consecutive images (~0.7 × 0.7 μm) taken from those parts of the recording showing that spot appearance is simultaneous. (**D** and **E**) Routes for loss of either the U4 or U5 fluorescent spots at 2 mM (**D**, N = 128) or 50 μM ATP (**E**, N = 207) in two typical experiments collected under otherwise identical conditions. Red and green shapes represent observation of fluorescence from the corresponding Dy549 (green-excited) or Cy5 (red-excited) fluorophores on U4 or U5, respectively; grey shapes represent the absence of fluorescence. Percentages represent the fraction of U4/U5 complexes in which fluorescence disappeared by the indicated pathway; more prevalent pathways are emphasized with thicker arrows.

The following figure supplements are available for figure 3:

**Figure supplement 1.** Additional examples (to supplement those shown in *Figure 3B*) of fluorescence intensity traces showing individual U5-DHFR subcomplexes (red) co-localizing with U4-SNAP subcomplexes (dark green) on surface-tethered pre-mRNA in the presence of 2 mM ATP.

**Figure supplement 2.** Additional examples (to supplement those shown in *Figure 3C*) of fluorescence intensity traces showing individual U5-DHFR subcomplexes (red) co-localizing with U4-SNAP subcomplexes (light green) on surface-tethered pre-mRNA in the presence of 50 μM ATP.

**Figure supplement 3.** Probability density histogram of dwell times (points) for U4 recorded at 150 (blue) and 450 μW (red) laser powers, and global fit to a two-exponential model that includes power-proportional photobleaching.

a U4 spot at the same location (e.g., *Figure 3B* and *Figure 3—figure supplement 1*). Similarly 46% of U4 spots (467 of 1016 total) arrived simultaneously with a U5 spot appearance. Colocalization of simultaneously arriving U4 and U5 spots sometimes persisted for tens or hundreds of seconds; these events likely reflect tri-snRNP binding. Consistent with this interpretation, simultaneous appearances of U4 and U5 at control areas of interest (AOIs) with no detected pre-mRNA molecules were extremely rare ($\sim 3 \times 10^{-4}$ events AOI$^{-1}$ min$^{-1}$). In contrast, the U4-only and U5-only binding events were typically transient, often lasting only a single frame (<3 s), and were not analyzed further (data not shown).

In contrast to the observations at 2 mM ATP, we observed at 50 µM ATP a higher proportion of coincident U5 and U4 binding [79% of U5 spots appeared simultaneously with a U4 spot (223/284) and 85% of U4 spots appeared simultaneously with a U5 spot (223/262); e.g., *Figure 3C* and *Figure 3—figure supplement 2*]. Because the same preparation of labeled U4/U5 extract was used in both 2 mM and 50 µM ATP experiments, it is probable that differences in the amount of coincident U4 and U5 binding reflect differences in tri-snRNP relative abundance under the two conditions; differences that likely originate from tri-snRNP disruption that accompanies spliceosome activation and splicing occurring at 2 mM ATP but not at 50 µM ATP. In addition, the tri-snRNP itself is destabilized by the addition of ATP, leading to an decrease in its abundance relative to free U4 and U5 snRNPs and the U4/U6 di-snRNP (*Raghunathan and Guthrie 1998a*; *1998b*; *Huang et al., 2014*).

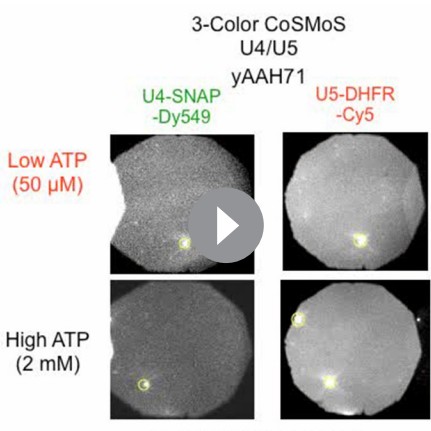

**Video 2.** Surface accumulation of the U4 and U5 snRNPs on surface-tethered pre-mRNAs (not shown) under both low (50 µM) and high (2 mM) ATP conditions. Spliceosome activation is prevented at low ATP. White spots represent fluorescent U4 or U5 subcomplexes labeled with Dy549-SNAP$_f$ tags or Cy5-TMP tags, respectively. This movie was obtained from recordings (60 min) of 1 s duration frames recorded at 3 s intervals. To reduce the movie file size, only every third frame was included from 0 to 45 min in the movie file, and the movie is played back at 20 frames/s. Each of the four recordings shows a $\sim$20 x 20 µm field of view, and the four individual movie files were combined using Apple Keynote software.

## ATP-dependent outcomes of tri-snRNP binding

Based on the idea that the U4 and U5 coincident arrivals in the three-color CoSMoS experiments represented tri-snRNP binding events leading to B complex formation, we selected just those events for further analysis. We observed four different outcomes from these coincident binding events (*Figure 3D,E*): (1) the U4 snRNP spot disappeared while the U5 spot remained; (2) the U5 snRNP spot disappeared while the U4 spot remained; (3) both the U4 and U5 spots disappeared simultaneously; or (4) both the U4 and U5 signals persisted on the pre-mRNA until the experiment was terminated after one hour. In the two cases in which one spot disappeared before the other, disappearance could be caused by photobleaching of the dye label, or by dissociation of the individual snRNP from B complex, leaving the other behind. However, simultaneous disappearance of both spots most likely represents U4/U6.U5 tri-snRNP dissociation because simultaneous photobleaching is improbable.

In the presence of 2 mM ATP, the predominant outcome (62%; 79/128 events in a given experiment) of coincident U4.U5 binding was U4 spot loss followed by U5 spot loss (*Figure 3D*). This outcome is consistent with the expected U4 loss accompanying B complex activation followed by U5 loss after splicing. Loss of the U5 signal prior to loss of U4 was rarely seen (2%), and in no case did the U4.U5 signals persist until the end of the experiment. The remainder mostly (36%) followed the route in which U4 and U5 spots disappeared simultaneously. These results are consistent with the majority of tri-snRNP interactions with pre-mRNA at 2 mM ATP resulting in activation and U4 dissociation. The smaller fraction with simultaneous spot disappearance may represent release of intact tri-snRNP without spliceosome activation.

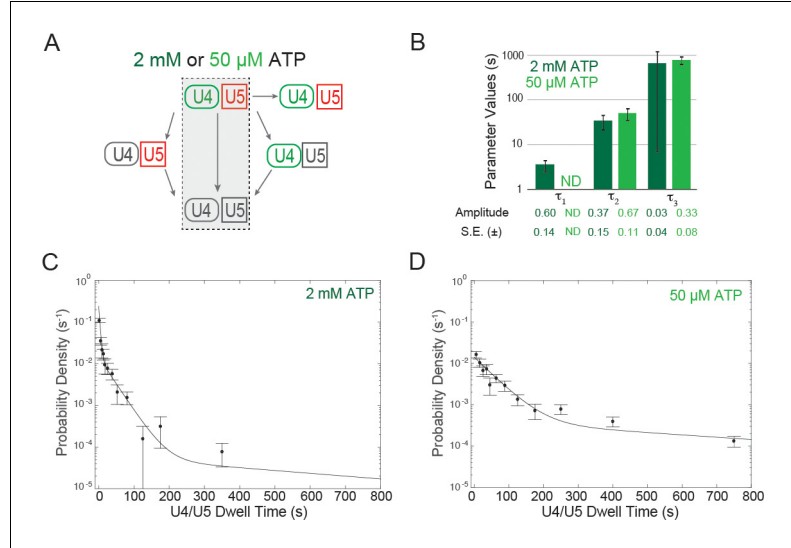

**Figure 4.** Dwell time analysis at 2 mM and 50 µM ATP of complexes containing both U4 and U5 for which the two snRNPs both arrived at and departed from the pre-mRNA simultaneously. (**A**) Routes for loss of snRNP fluorescence (as in *Figure 3D,E*); shading indicates a subset of events, interpreted as tri-snRNP dissociation, that are analyzed in this figure. (**B**) Fit parameters (± S.E.) describing the distribution of U4/U5 dwell times at 2 mM and 50 µM ATP (see also *Figure 4—source data 1*). The shortest dwell time component (here described as $\tau_1$) appeared absent at 50 µM ATP (ND, Not Determined). (**C, D**) Probability density histogram of U4/U5 dwell times observed at 2 mM ATP (**C**; N = 128 events on 471 pre-mRNAs) or 50 µM ATP (**D**; N = 164 events on 210 pre-mRNAs). Lines represents fits of the dwell time distributions with equations containing three (**C**) or two (**D**) exponential terms that yielded the parameters reported in (**B**).

The following source data and figure supplement are available for figure 4:

**Source data 1.** Fit parameters describing the distribution of dwell times observed for U4/U5 complexes that arrived and departed simultaneously

**Figure supplement 1.** ATP dependence of discard of U4.U5 complexes assembled at 50 µM ATP.

Flux through these pathways changed dramatically when the ATP concentration was reduced to 50 µM (*Figure 3E*). The U4-first loss pathway was rarely observed (9%; 19/207 events in a given experiment), consistent with activation being blocked at 50 µM ATP. Instead, U4 and U5 most frequently (76%) disappeared simultaneously, with a small fraction (9%) not disappearing at all but persisting until the end of the experiment. These results suggest that when activation is suppressed by reducing ATP, tri-snRNP mostly releases intact but on rare occasions participates in formation of a kinetically stable complex on the pre-mRNA. The persistent complexes may correspond to the stalled B complexes previously isolated in ensemble studies (*Fabrizio et al., 1989*; *Tarn et al., 1993*). Further, the observation that U4-first loss is infrequently observed at 50 µM ATP confirms that most of the more frequent U4-first loss events observed at 2 mM ATP are due to U4 snRNP dissociation, not photobleaching.

## Dynamics of tri-snRNP disappearance reveal new features of spliceosome proofreading and remodeling

We next analyzed the kinetic behavior of the complexes formed by simultaneous arrival of U4 and U5 – i.e., the tri-snRNP binding events – at 50 µM and 2 mM ATP.

First, we measured the U4/U5 dwell times for those events in which U4 and U5 left the pre-mRNA simultaneously (*Figure 4A*). These events presumably represent release of bound tri-snRNP without activation. At both 50 µM and 2 mM ATP the dwell time distributions were multi-exponential (*Figure 4B–D*), suggesting the presence of multiple distinct types of tri-snRNP/pre-mRNA complexes. At 50 µM ATP, the dwell time distribution fit to the sum of two exponential terms

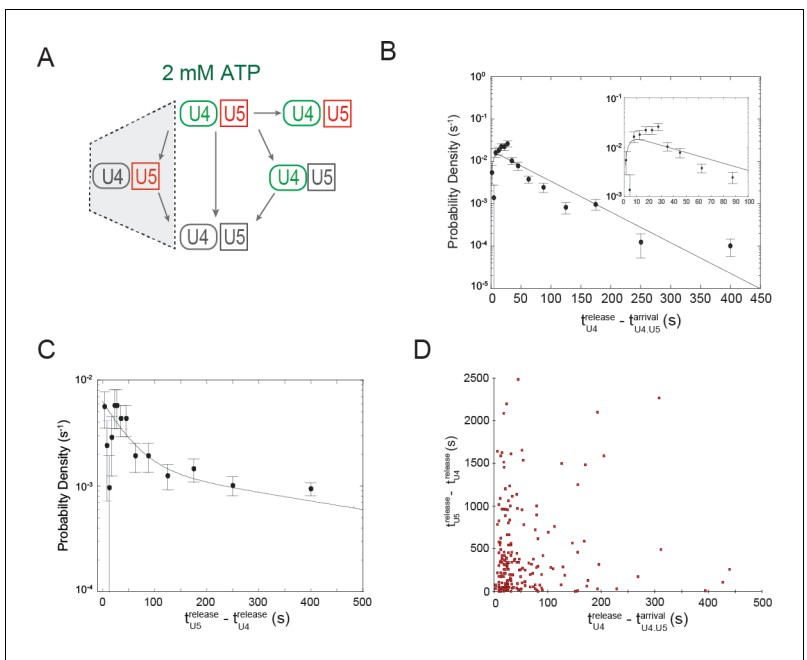

**Figure 5.** Dwell time analysis at 2 mM ATP of complexes containing both U4 and U5 for which the two snRNPs both arrived at the pre-mRNA simultaneously but in which U4 fluorescent spot disappearance preceded U5 spot disappearance. (A) Routes for loss of snRNP fluorescence (as in *Figure 3D,E*); shading indicates a subset of events, interpreted as spliceosome activation, that are analyzed in this figure. (B) Probability density histogram of dwell times for U4 molecules in the analyzed subset ($t_{U4}^{release} - t_{U4.U5}^{arrival}$; N = 244 events on 471 pre-mRNAs). The line represents a fit of the dwell time distribution to a convolution function. (C) Probability density histogram of dwell times for U5 molecules in the departure pathway subsequent to U4 departure ($t_{U5}^{release} - t_{U4}^{release}$; N = 207 events on 471 pre-mRNAs). The line represents a fit of the dwell time distribution to an equation containing two exponential terms. (D) The dwell time of each U5 ($t_{U5}^{release} - t_{U4}^{release}$) plotted against the dwell time of U4 ($t_{U4}^{release} - t_{U4.U5}^{arrival}$) in the same complex. Fit parameters for (C) and (D) are given in *Figure 5—source data 1*.

The following source data is available for figure 5:

**Source data 1.** Fit parameters describing the distribution of dwell times observed for U4 or U5 after simultaneous U4/U5 Arrival and U4 loss preceding U5 loss.

(*Figure 4D*), whereas the distribution at 2 mM ATP required three exponential terms for a satisfactory fit (*Figure 4C*). The two time constants at 50 µM ATP (52 and 780 s) appear to most closely correspond to the two longest time constants at 2 mM ATP (34 and 660 s; *Figure 4B*; *Figure 4— source data 1*). However, at 2 mM ATP the majority of dissociation events are not in these longer components but rather in the shortest-lived component (4 s). Thus, our observations suggest that at 2 mM ATP most non-productively bound tri-snRNPs are quickly discarded by a pathway not operative or much slower at 50 µM ATP.

To probe the ATP-dependence of this potential discard pathway further, we assembled stalled complexes on surface-tethered pre-mRNAs at 50 µM ATP. We then exchanged the WCE containing 50 µM ATP with splicing buffer, also containing 50 µM ATP, to remove any unstable tri-snRNP-containing complexes (*Figure 4—figure supplement 1A*). During this first buffer wash, the majority of U4.U5 complexes remained bound to the slide (10–17% decrease in U4.U5 spot density, N = 253 ± 18 U5 spots per field of view in WCE prior to the wash). We next carried out a second buffer exchange and flowed in buffer containing either 50 µM or 2 mM ATP. When a second 50 µM ATP wash was carried out, the majority of U4/U5 complexes remained bound (~19% decrease in spot density between the first and second wash steps, *Figure 4—figure supplement 1B*). In contrast, the majority of U4 and U5 spots were released when a 2 mM ATP wash was used: 74% and 68% decreases in U4 and U5 spot density, respectively, and only a small fraction of remaining U4 and U5

particles colocalized with one another (*Figure 4—figure supplement 1B–D*). Release of U4 and/or U5 occurred very rapidly—within the deadtime of our experiment (~30 s). Loss of signals from both U4 and U5 is most consistent with ATP-dependent discard of these complexes since U5 should be retained in B$^{act}$ spliceosomes (*Figure 1A*). Furthermore, these results suggest that the ATPase responsible for the discard of U4 and U5 is stably associated with the pre-mRNA and remains bound when the WCE is removed and replaced by buffer. Our data indicate that the majority (~68%) of stable U4.U5-containing complexes assembled at 50 μM ATP in WCE will be discarded in the presence of 2 mM ATP in buffer and only a smaller fraction (at most 32%) can potentially form B$^{act}$ spliceosomes.

We next analyzed the pathway that is dominant at 2 mM ATP: events in which U4 and U5 fluorescence appeared simultaneously but the U4 fluorescence was lost prior to U5 loss (*Figure 5A*). This cohort is expected to include the pre-mRNA molecules with productively bound tri-snRNPs that progress to spliceosomal activation and splicing. The U4 dwell time distribution for this activation cohort ($t_{U4}^{release} - t_{U4.U5}^{arrival}$; *Figure 5B*) is distinct from that in the non-productive cohort that release tri-snRNP (*Figure 4*); this demonstrates that the U4- and U5-containing complexes in the two cohorts must have different functional and structural properties even prior to dissociation. In the activation cohort, only a few U4 molecules left immediately after the tri-snRNP bound; indeed, the distribution of $t_{U4}^{release} - t_{U4.U5}^{arrival}$ values shows a dearth of short events (*Figure 5B* and *Figure 5—source data 1*). The distribution fit to a convolution of two exponentials, suggesting that U4 dissociation that accompanies activation is preceded by at least two sequential steps that are each partially rate limiting (*Colquhoun and Hawkes 1995*). This is not surprising, given that spliceosome activation encompasses multiple conformational transitions and points of regulation (*Brow 2002*; *Small et al., 2006*; *Brenner and Guthrie 2006*; *Bellare et al., 2008*). We also measured how long it took for each U5 spot to disappear after U4 disappearance ($t_{U5}^{release} - t_{U4}^{release}$); these U5-containing molecules include

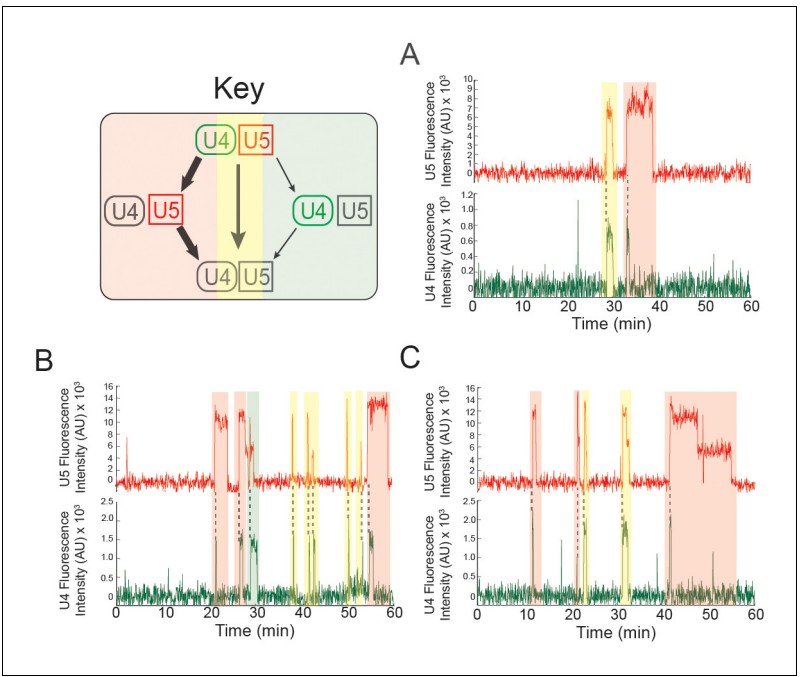

**Figure 6.** Examples of pre-mRNA molecules in which different outcomes result from individual tri-snRNP binding events at 2 mM ATP. Events are color-coded according to the pathways shown in the Key. The same pre-mRNA molecule may interact with the tri-snRNP by releasing U4 and U5 simultaneously or proceeding through activation (**A-C**). Some pre-mRNA molecules exhibited multiple rounds of activation (**B** and **C**).

The following figure supplement is available for figure 6:

**Figure supplement 1.** Analysis of multiple tri-snRNP binding events at 2mM ATP.

both on-pathway complexes that acquire NTC and subsequently splice, as well as those that are discarded prior to splicing completion. Consistent with the expected presence of multiple U5-containing species, the dwells fit to the sum of two exponentials (*Figure 5C* and *Figure 5—source data 1*).

RNAs can adopt multiple secondary structures, and it is possible that after tri-snRNP binding there exist multiple spliceosome configurations that differ in their behavior after activation. We tested this idea by asking whether ($t_{U4}^{release} - t_{U4.U5}^{arrival}$; a reporter of activation) was correlated with the following U5 dwell time ($t_{U5}^{release} - t_{U4}^{release}$; which reflects the speed of the subsequent processes). Because there was no clear correlation between the rate of U4 loss and the subsequent U5 dwell time (*Figure 5D*), there was no evidence for multiple configurations with differing intrinsic kinetics that persist through activation.

## Tri-snRNP activation and dissociation pathways can both occur on the same pre-mRNA molecules

Even under splicing conditions at 2 mM ATP, we often observed multiple sequential tri-snRNP binding and dissociation events on the same pre-mRNA molecule (~22% of pre-mRNAs showed multiple tri-snRNP binding events, *Figure 6—figure supplement 1A*). The fraction of pre-mRNAs that only showed one tri-snRNP binding event (~22%) is much lower than the fraction of RP51A substrates that typically splice in our assays (~60%, *Figure 2—figure supplement 1*), although splicing on slides may be lower. This suggests that at least some pre-mRNAs that do splice also bind the tri-snRNP more than once. When multiple tri-snRNP associations were observed they were nearly always sequential: a second tri-snRNP binding was very rarely seen unless both U4 and U5 fluorescence from the previous association had already disappeared (simultaneous colocalization of two tri-snRNPs was potentially observed on only 6 out of 481 RNAs, ≤1%). Some tri-snRNP binding events on a pre-mRNA molecule resulted in simultaneous appearance of both the U4 and U5 signals followed by their simultaneous loss, usually a short time later (*Figure 6*, yellow highlights). Other tri-snRNP binding events on the same pre-mRNA molecule would show simultaneous appearance of both U4 and U5 followed by loss of U4 fluorescence prior to loss of U5 (i.e., spliceosome activation; *Figure 6*, red highlights). Thus, a transcript is able to recruit the tri-snRNP multiple times during spliceosome assembly, but not every recruitment to a given pre-mRNA molecule leads to activation.

If loss of U4 snRNP from B complex commits the pre-mRNA to splicing, we would expect to see loss of U4 without simultaneous loss of U5 to occur only once on each pre-mRNA. Instead, we observed that approximately 10% (49 of 481) of pre-mRNAs under 2 mM ATP exhibited at least two repeats of U4/U5 simultaneous binding, followed by loss of the U4, followed by loss of U5 (*Figure 6B,C*). Since binding of the tri-snRNP is dependent on presence of an intron (*Shcherbakova et al., 2013*; *Hoskins et al., 2011*), we conclude that on these pre-mRNAs the first U4 release event did not result in productive splicing and that subsequent observations of U4 release result from spliceosome re-assembly on the same pre-mRNA. These data indicate that activation does not always commit the pre-mRNA to splicing, and are consistent with the presence of discard pathways that can prevent splicing subsequent to activation (*Mayas et al., 2006*; *Koodathingal et al., 2010*).

Since tri-snRNP association is reversible (*Figure 4* and [*Hoskins et al., 2011*]), it is not surprising that we observed tri-snRNP to re-associate multiple times with the same pre-mRNA. In contrast, reassociation of U4 complexes containing the labeled protein Prp3 after prior U4 departure was exceedingly rare. Of 260 cases in which both U4 and U5 fluorescence appeared simultaneously followed by disappearance of U4 and then U5, we observed only 15 instances (5.7%) of reappearance of U4 while U5 remained visible. It is likely that some of these rare events arose from background binding of fluorescent molecules to the slide surface since control experiments revealed that the rate of background U4 binding is similar to the observed rate of U4 re-binding subsequent to U5 loss ($2.3 \times 10^{-3}$ vs. $9.8 \times 10^{-3}$ events/min, respectively). In addition, only 5 cases were found in which the rebound U4 spot was stable enough to last more than a single frame. Taken together, these results are consistent with activation being largely irreversible, with only a slim likelihood (<2%) for rebinding of Prp3-containing U4 or U4/U6 snRNPs.

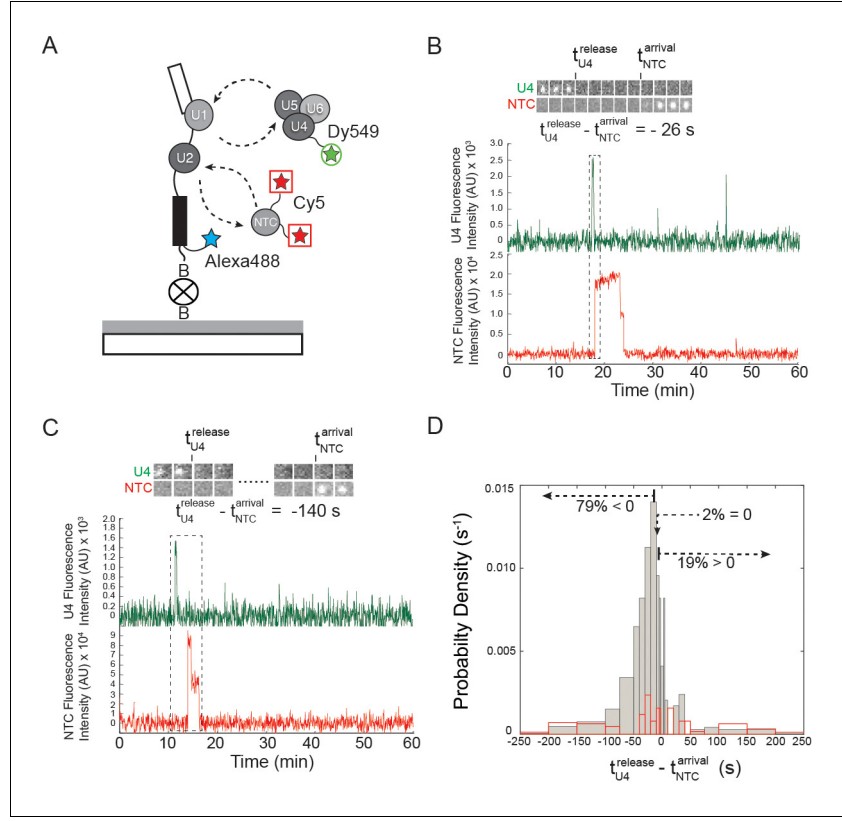

**Figure 7.** Three-color CoSMoS observation of U4 and NTC binding dynamics at 2 mM ATP. (**A**) Schematic of the experiment, in which the NTC was labeled with two red-excited (Cy5) fluorophores, U4 was labeled with a single green-excited (Dy549) fluorophore, and the surface-tethered pre-mRNA contained a single blue-excited (Alexa488) fluorophore. (**B** and **C**) Representative traces showing peaks in fluorescence intensity corresponding to colocalization of U4 (dark green) and NTC (red) with single pre-mRNA molecules. Raw images (~0.9×0.9 μm) corresponding to portions of the trace segments enclosed by the dashed boxes are included above the traces. The times of U4 release ($t_{U4}^{release}$) and NTC arrival ($t_{NTC}^{arrival}$) are determined from these data as shown. Subtraction of $t_{NTC}^{arrival}$ from $t_{U4}^{release}$ time yields a positive number if the NTC arrived prior to U4 release or a negative number if the NTC arrived after U4 release. (**D**) Probability density histogram showing the delay between NTC arrival and U4 loss (gray). The first and last bins (limits of -3600 – -200 and 200 – 3600 s, respectively) were truncated in the figure for clarity. Most often (79% of N = 293 total events on 402 pre-mRNAs), the NTC arrived soon after loss of the U4 signal ($t_{U4}^{release} - t_{NTC}^{arrival} < 0$). A randomized control histogram (see Materials and methods) is also shown (red). Comparison of gray and red curves suggests that U4 departed after NTC arrival in <19% of complexes.

The following figure supplement is available for figure 7:

**Figure supplement 1.** (A-E) Additional examples (to supplement those shown in *Figure 7*) of fluorescence intensity traces showing individual NTC-DHFR subcomplexes (red) co-localizing with U4-SNAP subcomplexes (dark green) on surface-tethered pre-mRNA in the presence of 2 mM ATP.

## Tri-snRNP rebinding after discard involves fewer steps than initial Tri-snRNP recruitment

We wondered if pre-mRNAs that bound the tri-snRNP more than once also recruited tri-snRNPs by different kinetic pathways. That is, do the kinetics of initial tri-snRNP binding to a pre-mRNA differ from the kinetics of tri-snRNP association after an activation attempt or discard? For each pre-mRNA, we measured the arrival time of the first tri-snRNP binding event ($t_{U4.U5}^{arrival}$) and plotted the distribution of arrival times in a histogram (*Figure 6—figure supplement 1B*). The histogram shows few events at small values of $t_{U4.U5}^{arrival}$ and could be described by a function that is a convolution of two exponential terms. This is consistent with multiple partially rate-limiting steps being required prior to

initial tri-snRNP binding to the pre-mRNA (*e.g.,* U1 and U2 binding (*Hoskins 2011*). Interestingly, the distributions of tri-snRNP arrival times following U4.U5 discard or U4 release events ($t_{U4.U5}^{arrival} - t_{U4.U5,discard}^{release}$ or $t_{U4.U5}^{arrival} - t_{U5,activation}^{release}$) both fit to distributions described by an equation with single exponential term and with similar fitted parameters (*Figure 6—figure supplement 1C,D*). This suggests that rebinding of the tri-snRNP to pre-mRNA following discard of tri-snRNPs or activated spliceosomes involves fewer steps than initial tri-snRNP recruitment, possibly because they are rebinding to an already partially assembled complex. Thus, splicing efficiency may potentially be maintained by not only multiple tri-snRNP binding events occurring on the pre-mRNA but also by facilitating subsequent recruitment of the tri-snRNP to those pre-mRNAs.

### NTC association typically occurs after U4 loss

The NTC is required for spliceosomal catalysis, but the timing of NTC association relative to other steps in activation is unresolved (*Hogg et al., 2010*). In order to address this issue, we carried out 3-color CoSMoS experiments in which the U4 snRNP was labeled with a green-excited fluorophore on Prp3 and red-excited fluorophores were attached to NTC components (Cef1 and Ntc90) with the DHFR tag (*Figure 7A*). This triply-tagged strain showed near-parental-strain growth kinetics and splicing activity (*Figure 2—figure supplement 1*). In 2 mM ATP, we observed transient U4 and longer-lasting NTC associations with pre-mRNA molecules, consistent with the behaviors seen in extracts in which U4 and NTC were individually labeled (*Figures 1* and *2*). We then measured for each pre-mRNA molecule the sequence of U4 and NTC appearance and disappearance (*Figure 7B, C* and *Figure 7—figure supplement 1*). In the majority of cases (79%), NTC signals appeared soon (typically ~30 s) *after* loss of U4 (*Figure 7D*). A randomized control (*Figure 7D*, red lines) indicates that this distribution did not arise by chance. In addition, the data show evidence for a small subpopulation (on order 10% of complexes) where NTC bound first but U4 release followed almost immediately (within a few seconds). Since ~53% of all molecules that recruit NTC subsequently are spliced (*Hoskins 2011*), the data suggest that the main pathway leading to functional spliceosomes is NTC association subsequent to U4 loss.

## Discussion

It has long been appreciated that activation represents an important crossroads for the spliceosome, separating assembly stages from catalysis. However, little is known about how activation is coordinated in terms of both the ordering of events and points of regulation. Here we demonstrate that the tri-snRNP can interact reversibly with pre-mRNA, that some tri-snRNP interactions do not result in release of U4, that when U4 is released this release is effectively irreversible, that ATP can promote both rapid activation and discard, and that NTC binding usually occurs only after U4 release. Together our data allow us to propose an integrated model for tri-snRNP association with the pre-spliceosome and subsequent spliceosome activation (*Figure 8*). The model proposes an ordered pathway for activation in which physiological ATP concentrations facilitate both U4 dissociation leading to NTC addition (progressing towards splicing) and tri-snRNP dissociation resulting in B-complex disassembly (progressing away from splicing).

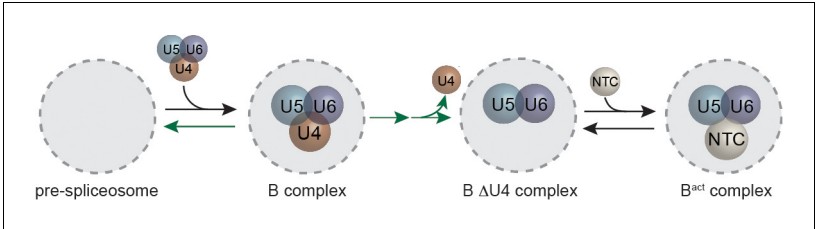

**Figure 8.** Model for tri-snRNP recruitment and spliceosome activation highlighting the predominant pathways observed in our experiments. The spliceosome and its other snRNP components are shown schematically as a grey circle. Additional observed kinetic components are not included in this simplified model. ATP stimulates both tri-snRNP discard and activation (green arrows). During activation, U4 loss occurs through a multi-step process and is irreversible. NTC joins the spliceosome after U4 release.

## Spliceosomes remain dynamic even when stalled by limiting ATP

Inhibiting spliceosomal activation by limiting ATP has been used in multiple laboratories to accumulate and purify B complex spliceosomes for analysis of their function and composition (*Chan et al., 2003*; *Tarn et al., 1993*; *Fabrizio et al., 2009*). It has often been assumed that the tri-snRNP binds to but cannot release from pre-mRNAs under these conditions. Our data show that this is not the case: the majority of tri-snRNP encounters with pre-mRNA molecules involved simultaneous U4 and U5 association and dissociation of both after just tens of seconds. While we did not examine U1 and U2 directly in these experiments, it is likely that these pre-mRNAs were assembled into pre-spliceosomes since single molecules of the tri-snRNP fail to accumulate on pre-mRNAs without ATP (*Hoskins et al., 2011*) and tri-snRNP association almost always requires prior U1 and U2 binding (*Shcherbakova et al., 2013*). Only a small fraction of events resulted in tri-snRNP retention at the conclusion of our experiments. Based on our data, it is unclear what distinguishes complexes that released the tri-snRNP from those in which it was retained for tens of minutes. It is likely that the longest-lived complexes are the assemblies that have been purified and characterized as the spliceosomal B complex (*Fabrizio et al., 2009*; *Tarn et al., 1993*). Our experiments (*Figure 4—figure supplement 1*) suggest that the majority of spliceosome complexes formed at 50 μM ATP and that survive removal of the WCE are not competent for $B^{act}$ formation and are instead rapidly discarded by an ATP-dependent mechanism. However, neither our work nor previous studies have directly established what fraction of the remaining long-lived complexes are kinetically competent intermediates on the splicing pathway at physiological ATP.

## Physiological ATP promotes both U4 and tri-snRNP release from B complex

Under limiting ATP conditions (*Figure 8*), the tri-snRNP associates with the pre-spliceosome but activation and release of U4 rarely occurs. Unexpectedly, at physiological ATP conditions that permit activation and splicing (*Figure 8*), tri-snRNP release is much faster than at 50 μM ATP. The reason that splicing can nevertheless occur is because at high ATP U4 release is rapid, irreversible, and is thus able to compete with tri-snRNP dissociation even though that dissociation rate has also increased. Based on the observed fluxes through the U4 release and tri-snRNP dissociation pathways, a significant fraction (one of every three) tri-snRNP binding events ends in tri-snRNP dissociation without spliceosome activation.

## Release of U4 is irreversible but does not guarantee splicing

While many steps in splicing have been shown to be readily reversible including snRNP and NTC association (*Hoskins et al., 2011*), pre-mRNA conformational changes (*Abelson et al., 2010*; *Krishnan et al., 2013*), and the chemical steps (*Tseng and Cheng 2008*), in our experiments the activation and loss of U4 appears to be almost irreversible. In >98% of events in which U4 was ejected after tri-snRNP binding, Prp3-containing U4 snRNPs did not reappear while U5 remained bound. This indicates effective irreversibility of U4 loss. Since U4 loss permits U6 to make both inter- and intramolecular basepairing interactions to form the spliceosome catalytic core and these are mutually exclusive with U4 basepairing, our results suggest that U4 is quickly recycled in the extract (*Raghunathan and Guthrie 1998a*) and/or that the conformational rearrangements within U6 occur rapidly, preventing re-base pairing with U4 directly or with the aid of the U4/U6 annealing protein Prp24.

Despite irreversible loss of U4 during the B to $B^{act}$ transition, spliceosomal activation does not guarantee that the pre-mRNA will be spliced by that particular spliceosome. We observed multiple instances in which a pre-mRNA underwent several rounds of tri-snRNP binding, U4 release, then U5 release and tri-snRNP rebinding (*Figure 6*). Thus, release of U4 during activation does not always commit a particular spliceosome to catalysis and mRNA release. This result is consistent with the presence of proofreading and discard steps carried out by Prp16 and Prp22 occurring after spliceosomal U4 release, and with disassembly of discarded spliceosomes by Prp43 (*Burgess and Guthrie 1993*; *Koodathingal et al., 2010*; *Mayas et al., 2006*). Our data support a model in which the spliceosome cannot reverse its steps back to the B complex if discard occurs after U4 release. Instead, U5 (and likely U6) are released and a new tri-snRNP must be recruited to the pre-mRNA for a new activation attempt.

Taken together our data suggest that tri-snRNP rebinding compensates for discard at the B complex or subsequent stages, contributing to the overall efficiency of splicing by permitting multiple attempts at assembly and activation.

## NTC association most frequently follows U4 release

The coordination of U4 release with other events during activation has not been well-studied. In particular, the relative timing of U4 release and NTC recruitment has remained ambiguous (*de Almeida and O'Keefe 2015*; *Hogg et al., 2010*). Early immunoprecipitation experiments suggested that the NTC joins the spliceosome soon after or concomitant with U4 loss (*Chan et al., 2003*; *Tarn et al., 1993*). In contrast, later mass spectrometry analyses of purified spliceosomes stalled by limiting ATP detected small amounts of NTC in B complex, suggesting that stable U4 and NTC binding are not mutually exclusive (*Fabrizio et al., 2009*). Our data explain this apparent contradiction by showing that at 2 mM ATP NTC recruitment soon after U4 release is the predominant pathway (79% of events). However, under limiting ATP (where there is little U4 release) transient interactions of NTC were still observed. More stable binding of NTC after U4 release is in agreement with both previous crosslinking results (*Rasche et al., 2012*) and more recent spliceosome structures (*Yan et al., 2015*) that show key binding sites for the NTC include the U6 internal stem loop (ISL) and the catalytic core. Because the ISL is not present when U6 is basepaired to U4, active site formation may be the major means by which the spliceosome recruits and retains the NTC.

In contrast with the predominant pathway, we less frequently (19% of events) observed NTC arrival prior to U4 loss. Recently it has been shown that functional tri-snRNPs can form in the absence of stable U4/U6 basepairing, suggesting that protein/RNA interactions can play a major role in tethering U4 to U5 and U6 (*Burke et al., 2015*). Thus, it is possible that the minor pathway we observe represents spliceosomes that have disrupted U4.U6 basepairing to facilitate NTC recruitment but in which other interactions have remained intact that delay U4 release.

Finally, at 50 µM ATP NTC failed to accumulate on the pre-mRNA molecules and instead displayed mostly transient interactions (*Figure 1*). This indicates that the NTC cannot stably bind spliceosomes stalled at B complex. The source of the small number of NTC peptides detected in mass spectrometry analyses of stalled B complex (*Fabrizio et al., 2009*) may reflect binding of the NTC to a small number of spliceosomes that may have been able to release U4 at 50 µM ATP (*Figure 3E*). Our single molecule data are consistent with data from human spliceosomes showing that NTC recruitment is also inhibited in complexes that fail to form B[act] when exchange of U1 for U6 pairing at the 5' SS is prevented (*Chiou et al., 2013*).

## Conclusion

Here, direct observation of binding and release of spliceosomal subcomplexes has allowed us to define the predominant pathways leading to spliceosomal activation (*Figure 8*). The results reveal that ATP stimulates both U4 release and tri-snRNP discard. ATP-dependent U4 release is known to be mediated by the Brr2 helicase (*Raghunathan and Guthrie 1998b*). The rapid tri-snRNP discard pathway is consistent with ATP-dependent proofreading by Prp28 during exchange of U1 for U6 basepairing at the 5' SS (*Yang et al., 2013*). If exchange is successful the spliceosome can proceed through the next steps of activation. If unsuccessful, Prp28 may induce rapid tri-snRNP release. It is possible that the discard pathway observed here may be related to prior observations of yeast pre-spliceosomes containing weakly bound tri-snRNPs upon inhibition of U1/U6 exchange (*Staley and Guthrie 1999*) and the labile tri-snRNPs present in human exon-defined splicing complexes (*Boesler et al., 2015*). The existence of this ATP-dependent tri-snRNP release pathway may serve to promote efficient splicing by preventing cellular accumulation of non-functional assemblies of snRNPs on pre-mRNAs.

By labeling protein components associated with U4 release, we have now demonstrated that all of the spliceosomal snRNPs and NTC are amenable to CoSMoS assays (*Rodgers et al., 2015*; *Hoskins et al., 2011*). Thus, there now exists a powerful toolset for studying compositional changes and dynamics of single spliceosomes in real time. Further mechanistic insights are likely to result from combining these tools with mutations in key spliceosomal components to better dissect the roles of individual proteins and snRNAs in the steps of spliceosomal assembly, activation, proofreading, chemistry, and recycling. Even in the absence of further studies, the data presented here define

the sequence of major molecular events in spliceosome activation and explain how splicing can be efficient despite high fluxes through discard pathways after tri-snRNP binding and after activation.

## Materials and methods

### Preparation of Pre-mRNAs

Capped RP51A pre-mRNA for in vitro splicing assays were transcribed in the presence of $[\alpha\text{-}^{32}P]UTP$ and purified as previously described (*Crawford et al., 2008*). Dye-labeled RP51A pre-mRNAs used in single molecule experiments were prepared by splinted ligation of a trace [$^{32}P$]-labeled, capped RP51A transcript to a biotinylated 2′-O-methyl oligonucleotide derivatized with a single Alexa Fluor 488 (Alexa488, ThermoFisher Scientific; Waltham, MA) or Alexa Fluor 647 (Alexa647, ThermoFisher Scientific; Waltham, MA) fluorophore as previously described (*Crawford et al., 2008*).

### Preparation of yeast strains

Yeast strains containing fast SNAP (SNAP$_f$) (*Sun et al., 2011*) tags on U4 snRNP proteins (*Table 1*) were prepared by homologous recombination as previously described (*Hoskins et al., 2011*; *Shcherbakova et al., 2013*). In brief, PCR products containing homology to the 3′ region of the PRP3 or PRP4 genes and downstream genomic DNA, a (Gly-Ser-Gly)$_2$ linker, the SNAP$_f$ tag, and a yeast selectable marker (hygromycin/HygR or nourseothricin/NatR) were generated using primers listed in *Figure 2—source data 1* and the plasmids pAAH0034 (containing the SNAP$_f$ gene and the NatR marker, [*Shcherbakova et al., 2013*]) or pAAH0013 (containing the SNAP$_f$ gene and the HygR marker). The pAAH0013 plasmid was prepared by cloning the SNAP$_f$ gene into the HindIII and BamHI sites of pAG32 (Euroscarf). The purified PCR products were then transformed into yeast using the lithium acetate method and colonies selected for growth in the presence of either hygromycin (300 µg/mL) or nourseothricin (100 µg/mL). SNAP$_f$ tags were confirmed by PCR using primers listed in *Figure 2—source data 1* as well as by labeling of the proteins with fluorophores (see below).

### Yeast growth rate analysis

Yeast growth rates were determined using a protocol adapted from Biotek (*Held 2010*). Experiments were carried out in a Tecan multimode plate reader by dilution of overnight cell cultures into flat-bottomed, clear, 96-well plates (200 µL, typically 8 replicates) and covering with optically clear TopSeal-A plate sealers. Shaking speed was set to slow with 1-mm amplitude, and plates were incubated at 30°C. Cell growth was monitored at 600 nm by recording measurements at ~ 2 min intervals. Growth rates were calculated as previously described (*Hoskins et al., 2011*) and compared to the parental control strain (yAAH0001).

### Preparation of yeast whole-cell splicing extracts

Yeast whole cell extract (WCE) was prepared as previously described (*Ansari and Schwer 1995*) with the following modifications. The concentration of DTT was increased to 1 mM at all steps. Lysates were aliquoted (1.25 mL), frozen in liquid $N_2$, and stored at -80°C after the second high speed spin. The lysate buffer was exchanged into 50 mM HEPES/KOH pH 7.9, 50 mM KCl, 10% (v/v) glycerol, and 1 mM DTT by gel filtration using a previously published protocol (*Anderson and Hoskins 2014*), aliquoted (42 µL), frozen in liquid $N_2$, and stored at -80°C. In cases where SNAP-tagged proteins were labeled, the lysate was incubated for 30 min at room temperature with the fluorophore (*e.g.*, benzylguanine-Dy549/SNAP-Surface 549, New England Biolabs; Ipswich, MA) prior to gel filtration. A fluorophore concentration of 2.1 µM was used to label SNAP tags, and fast SNAP (SNAP$_f$) tags were labeled using 1.1 µM fluorophore.

### Visualization of SNAP-tagged proteins in SDS-PAGE gels

SNAP-tagged proteins derivatized with fluorophores were visualized by denaturing polyacrylamide gel electrophoresis (SDS-PAGE) followed by imaging fluorescence on a Typhoon biomolecular imager (GE Life Sciences; Pittsburgh, PA). Data were analyzed using ImageQuant software. The extents and rates of fluorophore incorporation into SNAP and SNAP$_f$ tags were determined using previously described protocols (*Hoskins et al., 2011*).

## In vitro splicing assays

Splicing assays were carried out as previously described (*Crawford et al., 2008*) using 40% WCE and ~0.2 nM RP51A substrate. [$^{32}$P]-labeled RNAs were visualized by denaturing PAGE followed by phosphorimaging. Data were analyzed using ImageQuant software (GE Lifesciences; Pittsburgh, PA).

## Single-molecule experiments

Single molecule imaging chambers and buffers including oxygen scavengers and triplet quenchers were prepared as previously described (*Hoskins et al., 2011*; *Crawford et al., 2008*; *Anderson and Hoskins 2014*). Single-molecule data were collected on a homebuilt, micromirror TIRF microsope at room temperature as previously described (*Friedman et al., 2006*; *Hoskins et al., 2011*). To facilitate correction of stage drift, chambers also included fluorescent beads (TransFluoSpheres, streptavidin-labeled, 40 nm, 488/645) to act as fiducial markers. Laser powers were typically set at either 150 or 450 µW for both the 532 and 633 nm lasers. The distributions of dwell times from both laser powers in experiments at 2 mM ATP were compared to assess the impact of photobleaching and showed little difference in the fitted parameters (*Figure 3—figure supplement 3*). In three-color CoSMoS experiments, the 488 nm laser was set to 800 or 1750 µW and RNAs containing 488-excitable fluorophores were only imaged at the beginning and end of the experiment to avoid photobleaching of other fluorophores. In three-color CoSMoS experiments, data were collected with a frame duration of 1 s, a spacing between frames of 3 s, and with simultaneous illumination with both the 532 and 633 nm lasers. The same frame duration and spacing was used to collect data for two-color CoSMoS experiments following U4 binding dynamics except that only the 532 nm laser was used. Two-color CoSMoS experiments of U1 and NTC binding dynamics under low ATP used a frame duration of 1 s, a spacing between frames of 5 s, and illumination with the 532 nm laser. In all experiments, autofocusing was carried out once per minute with a 785 nm laser (*Hoskins et al., 2011*). A reference data set used for mapping the <635 nm and >635 nm fluorescence emission fields of view onto one another was acquired with each experiment (*Hoskins et al., 2011*).

## Data analysis

Data was analyzed essentially as described (*Hoskins et al., 2011*) by mapping the fields of view onto one another, correcting for stage drift, determining locations of immobilized pre-mRNAs (areas of interest, AOIs), and integrating pixel intensity at each AOI using custom Matlab software (The Mathworks; Natick, MA; https://github.com/gelles-brandeis). Peaks in intensity were identified by changes occurring more than 3.2σ over the baseline noise, and signals ended when they fell below 1σ. Each peak was manually inspected to confirm the presence of a spot of fluorescence centered in the AOI.

The distributions of observed dwell times for each subcomplex were displayed by constructing probability density plots in which the dwell times were binned and each bin divided by the product of the bin width and total number of events. Error bars for each bin were calculated as the error of a binomial distribution as previously described (*Hoskins et al., 2011*). Distributions described by one or two exponential terms were fit by maximum likelihood methods to exponential probability density functions as previously described for single and double exponential distributions (*Equations 1* or *2*, respectively; *Hoskins et al., 2011*). Distributions described by three exponential terms or by a convolution function (*Lu et al., 1998*) were fit to *Equations 3* or *4*, respectively. In all equations, $t_m$ represents the minimum detectable dwell time; $t_{max}$ represents the duration of the experiment (60 min); $A_1$ and $A_2$ the fitted amplitudes; and $\tau_1, \tau_2, \tau_3$ represent the fitted parameters. Errors in the fit parameter were determined by bootstrapping 1000 random samples of the data and determining the standard deviation of the resultant values.

$$\left[\left(A_1 \cdot \left(e^{-\frac{t_m}{\tau_1}} - e^{-\frac{t_{max}}{\tau_1}}\right)\right)\right]^{-1} \cdot \left[\frac{A_1}{\tau_1} e^{\frac{-t}{\tau_1}}\right] \tag{1}$$

$$\left[\left(A_1 \cdot \left(e^{-\frac{t_m}{\tau_1}} - e^{-\frac{t_{max}}{\tau_1}}\right)\right) + \left((1-A_1) \cdot \left(e^{-\frac{t_m}{\tau_2}} - e^{-\frac{t_{max}}{\tau_2}}\right)\right)\right]^{-1} \cdot \left[\frac{A_1}{\tau_1} e^{\frac{-t}{\tau_1}} + \frac{1-A_1}{\tau_2} e^{\frac{-t}{\tau_2}}\right] \tag{2}$$

$$\left[ \left( A_1 \cdot \left( e^{-\frac{t_m}{\tau_1}} - e^{-\frac{t_{max}}{\tau_1}} \right) \right) + \left( A_2 \cdot \left( e^{-\frac{t_m}{\tau_2}} - e^{-\frac{t_{max}}{\tau_2}} \right) \right) + \left( (1 - A_1 - A_2) \cdot \left( e^{-\frac{t_m}{\tau_3}} - e^{-\frac{t_{max}}{\tau_3}} \right) \right) \right]^{-1} \cdot$$
$$\left[ \frac{A_1}{\tau_1} e^{\frac{-t}{\tau_1}} + \frac{A_2}{\tau_2} e^{\frac{-t}{\tau_2}} + \frac{1 - A_1 - A_2}{\tau_3} e^{\frac{-t}{\tau_3}} \right] \tag{3}$$

$$\left[ (1/\tau_1 \cdot 1/\tau_2) \cdot \left( e^{-t/\tau_1} - e^{-t/\tau_2} \right) \right] \cdot \left[ \left( 1/\tau_2 \cdot \left( e^{-t_m/\tau_1} - e^{-t_{max}/\tau_1} \right) \right) - \left( 1/\tau_1 \cdot \left( e^{-t_m/\tau_2} - e^{-t_{max}/\tau_2} \right) \right) \right]^{-1} \tag{4}$$

## Acknowledgements

We thank members of the Gelles, Moore, Hoskins, Brow, and Butcher laboratories for helpful discussions. MJM is an Investigator of the Howard Hughes Medical Institute and supported by NIH R01 GM50037. JG and LF are supported by NIH R01 GM081648. AHH acknowledges support from startup funding from the University of Wisconsin-Madison, Wisconsin Alumni Research Foundation (WARF), and the Department of Biochemistry, awards from the National Institutes of Health (R00 GM086471, R01 GM112735), the Arnold and Mabel Beckman Foundation, and the Shaw Scientist Program of the Greater Milwaukee Foundation. MLR is supported by the Molecular Biophysics Training Program (NIH T32-GM08293).

## Additional information

### Funding

| Funder | Grant reference number | Author |
|---|---|---|
| National Institutes of Health | R01 GM50037 | Melissa J Moore |
| Arnold and Mabel Beckman Foundation | Beckman Young Investigator | Aaron A Hoskins |
| Greater Milwaukee Foundation | Shaw Scientist Award | Aaron A Hoskins |
| Howard Hughes Medical Institute | HHMI Investigator | Melissa J Moore |
| National Institutes of Health | R00 GM086471 | Aaron A Hoskins Margaret L Rodgers |
| National Institutes of Health | R01 GM112735 | Aaron A Hoskins Margaret L Rodgers |
| National Institutes of Health | R01 GM081648 | Larry J Friedman Jeff Gelles |
| National Institutes of Health | T32 GM08293 | Margaret L Rodgers |

The funders had no role in study design, data collection and interpretation, or the decision to submit the work for publication.

### Author contributions

AAH, Conception and design, Acquisition of data, Analysis and interpretation of data, Drafting or revising the article; MLR, LJF, Acquisition of data, Analysis and interpretation of data; JG, MJM, Analysis and interpretation of data, Drafting or revising the article

### Author ORCIDs

Aaron A Hoskins, http://orcid.org/0000-0002-9777-519X
Larry J Friedman, http://orcid.org/0000-0003-4946-8731
Jeff Gelles, http://orcid.org/0000-0001-7910-3421

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
