## [Decision Letter]

Thank you for submitting your article "Single Molecule Analysis Reveals Reversible and Irreversible Steps During Spliceosome Activation" for consideration by *eLife*. Your article has been favorably evaluated by Jean Beggs and Jonathan Staley, who is a member of our Board of Reviewing Editors, and the evaluation was overseen by Michael Marletta as the Senior editor.

The reviewers have discussed the reviews with one another and the Reviewing Editor has drafted this decision to help you prepare a revised submission.

Summary:

In this manuscript, Hoskins et al. exploit the unique and innovative capabilities of CoSMoS to study at unprecedented resolution the activation step of the spliceosome, arguably the most dramatic and catalytically-critical stage of spliceosome assembly. They investigate formation of the B complex and its conversion to the Bact complex by observing splicing substrate association and dissociation of U1, the tri-snRNP components U4 and U5, and/or the NTC, the complex characteristic of catalytically-activated spliceosomes. At low ATP, which stalls spliceosome assembly, the authors observed increased lifetimes of U1, U4, and U5 but decreased lifetimes of the NTC, consistent with previous observations. By simultaneously monitoring U4 and NTC binding and release from the substrate by three-color CoSMoS, the authors demonstrate that the NTC only transiently samples the substrate before U4 release and generally binds stably only after release, providing the first direct evidence for the dependence of stable NTC binding on U4 release. By co-localization of U4 and U5 by three-color CoSMoS, the authors also observed a substantial frequency of events in which both dissociate simultaneously. Surprisingly, at high ATP, there is a very fast phase of this event that predominates, implying a novel discard pathway that dominates the spliceosome activation phase. Importantly, the authors show that such discard events can be followed by rebinding of the tri-snRNP components. Further, while some studies on the U4/U6 annealing factor Prp24 have suggested action on the spliceosome, the authors provide the first evidence that U4 release from U6 on the spliceosome is irreversible and, strikingly, that subsequent U5 release can be followed by simultaneous reassociation of U4 and U5, revealing for the first time spliceosome re-assembly after the presumed discard of the splicing substrate and concomitant disassembly of the spliceosome.

Overall, the presentation is clear and enjoyable to read, the quality of the work is high, and the results are convincing. The analysis has not only provided direct evidence for previously suspected features of spliceosome assembly (e.g., the irreversibility of U4 release, the dependence of NTC binding on U4 release) but also revealed unexpected evidence for an apparently ATP-dependent discard pathway for the tri-snRNP and provided the first evidence for repeated cycles of tri-snRNP sampling of a substrate – both with and without U4 release.

Additionally, while the approach of CoSMoS has been previously introduced by the authors, the successful and revealing application here to the activation step in splicing will underscore the power of this approach in investigating factor dynamics to the broader community – well beyond the splicing field.

Essential revisions:

1) A main question here is what underlies the apparent ATP-dependence of U4/U5(U6) dissociation. Does the fast phase at high ATP reflect a direct role for ATP in discard, as the authors imply, or do differences in assembly at high ATP lead to a spliceosome in which the tri-snRNP dissociates more rapidly but without an ATP-dependence? To test directly whether the simultaneous release of U4 and U5 is ATP dependent, could the authors easily wash away (low) ATP in the middle of an experiment, after assembly on substrate, and assay for a decrease in the rate of U4/U5 dissociation? If not, minimally, this ambiguity should be acknowledged in the manuscript.

2) It seems that the data shown in Figure 6 represent a subset of the data from the experiment shown in Figure 3. What proportion of the RNAs undergo multiple tri-snRNP binding events?

3) Related to the above, can the authors provide any data in support of molecules that undergo multiple tri-snRNP binding events eventually proceeding to a functional splicing reaction, rather than being eventually discarded? In other words, how likely is it that such multiple association/dissociation events reflect a normal spliceosome assembly pathway? For example, is it possible to know which molecules become spliced? Could the authors simultaneously monitor release of the excised intron? Minimally, does the proportion of molecules that display multiple tri-snRNP binding events match an estimate of the number of molecules that undergo splicing in this setup? As is, the last line of the Abstract is not justified.

4) In the subsection “Spliceosomes remain dynamic even when stalled by limiting ATP” the authors discuss the need for binding of U1 and U2 prior to tri-snRNP binding. It should also be discussed, either here or later, that when tri-snRNP dissociates, these data do not appear to allow distinguishing between dissociation only of the tri-snRNP to leave a stable pre-spliceosome, or if U2 and/or U1 dissociate also, i.e. discard of the entire complex, and then assembly of a new spliceosome from scratch. If tri-snRNP discard leads back to the prespliceosome, then one would expect that tri-snRNP rebinding a second time would be kinetically faster than initial binding events. Can this be evaluated? (Similarly, in cases where U4 release is followed by U5 release and re-binding of the tri-snRNP, is rebinding faster than initial binding?)

---

## [Author Response]

*Essential revisions:*

1) A main question here is what underlies the apparent ATP-dependence of U4/U5(U6) dissociation. Does the fast phase at high ATP reflect a direct role for ATP in discard, as the authors imply, or do differences in assembly at high ATP lead to a spliceosome in which the tri-snRNP dissociates more rapidly but without an ATP-dependence? To test directly whether the simultaneous release of U4 and U5 is ATP dependent, could the authors easily wash away (low) ATP in the middle of an experiment, after assembly on substrate, and assay for a decrease in the rate of U4/U5 dissociation? If not, minimally, this ambiguity should be acknowledged in the manuscript.

We have carried out new experiments to test whether or not high ATP promotes rapid tri-snRNP dissociation by incorporating the reviewers’ suggestion and have found that 2 mM ATP in buffer is able to stimulate rapid release of U4 and U5 from complexes assembled at low (50 µM) ATP. Release occurs within the deadtime of the buffer exchange in our assay (~30s). Since both U4 and U5 are released, this is consistent with discard of these complexes. These experiments and the results are described in new text in the subsection “Dynamics of tri-snRNP Disappearance Reveal New Features of Spliceosome Proofreading and Remodeling” and in new Figure 4—figure supplement 1.

Together, these results provide additional evidence for ATP dependence of U4 and U5 release from pre-mRNA and indicates that the ATPase responsible is part of the complex that assembles at 50 µM ATP on pre-mRNA and remains bound following an initial 50 µM wash step. Furthermore, our results suggest that the majority (~68%) of complexes assembled at low ATP are discarded at high ATP and the maximum fraction of assembled complexes containing U4 and U5 that can continue on to form B^act^ spliceosomes is 32%.

2) It seems that the data shown in Figure 6 represent a subset of the data from the experiment shown in Figure 3. What proportion of the RNAs undergo multiple tri-snRNP binding events?

We have analyzed how many tri-snRNP binding events occur on each pre-mRNA at 2 mM ATP. Approximately 22% of pre-mRNAs undergo more than one tri-snRNP binding event. A histogram showing the number of tri-snRNP binding events observed on each pre-mRNA is reported in new Figure 6—figure supplement 1 and described in the subsection “Tri-snRNP activation and dissociation pathways can both occur on the same pre-mRNA Molecules”.

3) Related to the above, can the authors provide any data in support of molecules that undergo multiple tri-snRNP binding events eventually proceeding to a functional splicing reaction, rather than being eventually discarded? In other words, how likely is it that such multiple association/dissociation events reflect a normal spliceosome assembly pathway? For example, is it possible to know which molecules become spliced? Could the authors simultaneously monitor release of the excised intron? Minimally, does the proportion of molecules that display multiple tri-snRNP binding events match an estimate of the number of molecules that undergo splicing in this setup? As is, the last line of the Abstract is not justified.

We have not simultaneously monitored intron release; however, the number of molecules undergoing only 1 tri-snRNP binding event (~22%) is too small to account for the estimate of the number of molecules undergoing splicing (~60% of pre-mRNAs are able to go through at least the first step of splicing, as approximated from in vitro splicing assays in Figure 2—figure supplement 1). Therefore, these data strongly suggest that many pre-mRNAs that do splice also experience multiple rounds of tri-snRNP binding and release. We have mentioned this observation in the subsection “Tri-snRNP activation and dissociation pathways can both occur on the same pre-mRNA Molecules”. We have also replaced the verb “can” with “may” in the last sentence of the Abstract.

4) In the subsection “Spliceosomes remain dynamic even when stalled by limiting ATP” the authors discuss the need for binding of U1 and U2 prior to tri-snRNP binding. It should also be discussed, either here or later, that when tri-snRNP dissociates, these data do not appear to allow distinguishing between dissociation only of the tri-snRNP to leave a stable pre-spliceosome, or if U2 and/or U1 dissociate also, i.e. discard of the entire complex, and then assembly of a new spliceosome from scratch. If tri-snRNP discard leads back to the prespliceosome, then one would expect that tri-snRNP rebinding a second time would be kinetically faster than initial binding events. Can this be evaluated? (Similarly, in cases where U4 release is followed by U5 release and re-binding of the tri-snRNP, is rebinding faster than initial binding?)

The reviewers bring up an excellent point. To evaluate this we compared the time it takes for each pre-mRNA to bind the tri-snRNP initially compared with the times in between tri-snRNP binding events. For this latter subset of data, we further discriminated by times between tri-snRNP discard (simultaneous release of U4 and U5) and binding of the next tri-snRNP and by times between U4 release (i.e., B^act^ formation) and binding of the next tri-snRNP. The results of these kinetic analyses are shown in new Figure 6—figure supplement 1.

In brief, binding of the initial tri-snRNP shows different kinetic features than re-binding of the tri-snRNP after discard or U4 release. These data and results are described in the subsection “Tri-snRNP Rebinding After Discard Involves Fewer Steps than Initial Tri-snRNP Recruitment”. This suggests that some component of the system is “primed” for rebinding of the tri-snRNP as well as another mechanism for maintaining splicing efficiency, just as the reviewers propose.